# Cannabinoid Control of Olfactory Processes: The *Where* Matters

**DOI:** 10.3390/genes11040431

**Published:** 2020-04-16

**Authors:** Geoffrey Terral, Giovanni Marsicano, Pedro Grandes, Edgar Soria-Gómez

**Affiliations:** 1INSERM, U1215 NeuroCentre Magendie, 146 rue Léo Saignat, CEDEX, 33077 Bordeaux, France; geoffreyterral@gmail.com (G.T.); giovanni.marsicano@inserm.fr (G.M.); 2University of Bordeaux, 146 rue Léo Saignat, 33000 Bordeaux, France; 3Interdisciplinary Institute for Neuroscience, CNRS, UMR 5297, 33000 Bordeaux, France; 4Department of Neurosciences, University of the Basque Country UPV/EHU, Barrio Sarriena s\n, 48940 Leioa, Spain; pedro.grandes@ehu.eus; 5Achucarro Basque Center for Neuroscience, Science Park of the UPV/EHU, 48940 Leioa, Spain; 6IKERBASQUE, Basque Foundation for Science, Maria Diaz de Haro 3, 48013 Bilbao, Spain

**Keywords:** olfaction, endocannabinoids, olfactory epithelium, olfactory bulb, piriform cortex, CB1 receptor

## Abstract

Olfaction has a direct influence on behavior and cognitive processes. There are different neuromodulatory systems in olfactory circuits that control the sensory information flowing through the rest of the brain. The presence of the cannabinoid type-1 (CB1) receptor, (the main cannabinoid receptor in the brain), has been shown for more than 20 years in different brain olfactory areas. However, only over the last decade have we started to know the specific cellular mechanisms that link cannabinoid signaling to olfactory processing and the control of behavior. In this review, we aim to summarize and discuss our current knowledge about the presence of CB1 receptors, and the function of the endocannabinoid system in the regulation of different olfactory brain circuits and related behaviors.

## 1. The Endocannabinoid System: A General Overview

Cannabis sativa, also known as marijuana or cannabis, has been used for thousands of years for its therapeutic and recreational properties. Nowadays, after tobacco and alcohol, cannabis is the most commonly consumed drug of abuse, with 188 million cannabis users estimated worldwide in 2017 [1]. The cannabinoid receptors type-1 (CB1) and type-2 (CB2), their endogenous ligands (endocannabinoids), and the synthetic and degradative enzymes that regulate endocannabinoid levels support the concept of the endocannabinoid system (ECS) as participating in the regulation of physiological processes [2]. CB1 and CB2 receptors belong to the superfamily of G-protein-coupled receptors (GPCRs) that consist of seven transmembrane domains with an extracellular N-terminal and an intracellular C-terminal tail [3]. At the synaptic level, endocannabinoids can be synthesized, but not exclusively [4], by post-synaptic intracellular calcium elevations, which can be caused by various stimuli, including depolarization, the activation of metabotropic acetylcholine, and glutamate receptors, particularly Gq-coupled receptors (i.e., M1/M3 and mGluR 1/5) [3]. Once produced, endocannabinoids act on CB1 receptors that are mainly described at pre-synaptic terminals [2,3,5], and other cellular locations [6].

In neurons, the main effect of CB1 receptor activation is a decrease in neurotransmitter release, inducing different forms of endocannabinoid-mediated plasticity [3], such as the depolarization-induced suppression of inhibition/excitation (DSI/DSE; [7,8,9]), or the long-term depression of inhibitory/excitatory synapses [10,11,12,13,14]. CB1 receptors are widely expressed in the central nervous system and likely represent the most abundant GPCR in the brain [15]. Given its ubiquitous expression in multiple brain areas, CB1 receptors modulate a variety of functions, from sensory perception to more complex cognitive processes such as learning and memory [16,17,18].

## 2. Role of the Endocannabinoid System in Olfactory Circuits

Known for a long time, one of the predominant subjective effects of cannabis intoxication is the alteration of sensory perception, including olfactory processes [19]. However, although relatively high levels of CB1 receptors were described in the 1990s in many olfactory brain areas of rodents [20,21,22,23], their olfactory-related functions only started to be studied during the last decade. Notably, the involvement of CB1 receptors in specific odor-related processes has been reported in specialized olfactory structures such as the olfactory epithelium (OE; [24,25,26,27]), the main olfactory bulb (MOB; [18,28,29,30,31,32,33,34]), and the piriform cortex (PC; [35,36,37,38,39,40]), but also in other brain areas processing olfactory information [41,42,43,44]. For the sake of clarity in this review, we will focus on describing the role of the ECS, particularly CB1 receptor signaling, in specific main olfactory areas (i.e., OE, MOB, and PC).

## 3. The Endocannabinoid System in the Olfactory Epithelium

The first hypothesis for the physiological involvement of endocannabinoids in olfactory processes came from three observations: (1) The olfactory perception was shown to be changed depending on the feeding state of individuals [45,46], (2) the ECS were proposed to be involved in food intake [18,47], and (3) the anatomical and functional connectivity between peripheral organs regulating energy balance and olfactory structures [18,48]. Czesnik, Breunig, and colleagues [24,25] provided the first evidence that cannabinoids could modulate olfaction. These studies revealed the presence of CB1 receptors in the olfactory sensory neurons (OSN) of Xenopus laevis and demonstrated that endocannabinoids modulate odor-evoked responses. Additionally, they found that the production of endocannabinoids depends on the hunger state of the animal, which is responsible for changes in odor sensitivity activity. Similarly, CB1 receptors were also found in the OSN of rodents [27]. The CB1 receptor agonists changed odorant-induced cellular activity, but the authors did not observe olfactory consequences in the behavior of mutant mice lacking CB1 receptors (CB1-KO; [27]). Despite the species differences, several divergences appear between these studies. For instance, the first two studies evaluated the impact of cannabinoids on odor sensitivity by recording the cellular activity of the OSN with calcium imaging and electrophysiological methods [24,25]. Instead, Hutch and colleagues [27] investigated the involvement of CB1 receptors in olfactory-mediated learning and memory tasks such as the buried food test and a habituation/dishabituation paradigm. In addition, CB1-KO mice lack brain specificity and might be confounded by compensatory mechanisms [49]. Thus, the physiological role of CB1 receptors in the mammalian OE still remains unclear and will need further investigation.

## 4. The Endocannabinoid System in the Olfactory Bulb

In the mammalian MOB, the ECS was first described as a modulator of GABAergic transmission [28,33]. Pharmacological approaches, combined with in vitro patch-clamp experiments, highlighted that CB1 receptors modulate the firing pattern of periglomerular (PG) and external tufted cells (eTCs). Considering that PG cells form synapses with mitral and tufted cells [50], CB1 signaling may indirectly regulate the main output activity of the MOB neurons. Indeed, the inhibitory inputs of eTCs display spontaneous DSI [33], and the pharmacological manipulation of CB1 receptor signaling modulates mitral cell activity, likely through indirect control of inhibitory transmission [34]. These results suggest that endocannabinoids are capable of controlling mitral/tufted cell activity through the CB1 receptors on PG cells. Although the authors did not investigate the behavioral impact of these effects, the CB1 receptors’ activation may increase the signal-to-noise ratio and, thus, the overall sensitivity of the glomerulus to sensory inputs. Moreover, CB1 receptors are present in glutamatergic corticofugal fibers (CFF) coming from projection neurons from anterior cortical olfactory areas (including the anterior olfactory nucleus, AON, and the anterior piriform cortex), and targeting granule cells (GCs) of the MOB [32]. Consistent with the idea that cannabinoid signaling in the olfactory system might control the feeding state of the organism, the hypophagic phenotype observed in mice lacking CB1 receptors in their glutamatergic neurons is associated with an increased activity of CFF onto GCs. Notably, endocannabinoid levels increase in the MOB during fasting, allowing for the dampening of the excitation of GCs. Given that GCs control mitral cell activity, CB1 receptor activation of CFF induces the disinhibition of mitral cells. This effect is followed by a fasting-related enhancement in olfactory sensitivity, which correlates with the amount of food ingested upon refeeding. These results suggest that the endocannabinoid-mediated regulation of olfactory output information controls olfactory perception and food intake [32]. Since CB1 receptors have been described as being expressed on CFF fibers, they may thus regulate all of the downstream synapses of these fibers. This hypothesis was recently verified in the synapse between the CFF and the so-called deep short axon cells (dSAs; [31]). Indeed, depolarization of dSAs in the MOB of mice elicits pre-synaptic CB1 receptors’ transient suppression of excitatory CFF inputs (DSE). In addition, the authors demonstrated that dSAs could inhibit GCs, thereby suppressing GC to mitral cells inhibition. Interestingly, depending on the CFF synaptic strength, the CB1 receptor signaling can either control the synapses from dSAs to GCs, or directly from GCs to mitral cells, suggesting a double dissociation in the control of olfactory bulb output neurons [31]. However, the behavioral consequences of this bidirectional effect remain to be elucidated.

## 5. The Endocannabinoid System in the Piriform Cortex

The PC is a brain area capable of generating epileptiform activity [51]. In other brain structures such as the hippocampus, CB1 receptors have been shown to protect against seizures [52,53]. Thus, the anticonvulsant effects of cannabinoids were assessed in PC slices [36]. The authors demonstrated that CB1 receptor agonists reduce seizures, indicating that CB1 receptor activation is able to control PC activity [36]. However, there is currently no functional evidence about how the ECS could affect olfactory processes under pathological conditions such as epilepsy. Furthermore, the ECS in the PC indirectly affects social behavior [38]. Although it does not affect social interactions per se, local injections of a CB1 receptor antagonist into the posterior PC (pPC) reversed the impairment of social sniffing time induced by an activation of dopamine receptors, suggesting that the ECS in the pPC has a deleterious effect on social behavior when coupled with dopamine activation [38]. Moreover, the PC is an important area involved in olfactory memory [54]. Considering that the ECS is highly studied in learning and memory functions [16], other studies investigated its role in PC-dependent olfactory learning and memory. In the pPC, odor-discrimination training leads to the endocannabinoid-mediated modification of inhibitory synapses [35]. Indeed, the learning of a complex olfactory rule induces the activation of CB1 receptors, which in turn enhances GABAergic conductance in post-synaptic pPC pyramidal neurons, indicating a postsynaptic effect [35]. Despite the possible post-synaptic CB1 receptors’ localization, or that endocannabinoids can modulate directly postsynaptic GABAergic receptors [55], further experiments will determine how CB1 receptor activation allows for controlling GABAergic conductance in the pPC. In the anterior PC (aPC), CB1 receptors were mainly described at GABAergic synapses, where they modulate inhibitory transmission and plasticity [37,40]. Moreover, depending on CB1 receptors in the aPC, the retrieval of appetitive, but not aversive, olfactory memory is associated with a modulation of local inhibitory transmission onto specific principal cells in the aPC [37]. These data indicate that CB1 receptors in the aPC selectively control olfactory memory retrieval related to positively motivated behaviors. Thus, it will be crucial to determine if cannabinoid signaling controls functional connection between the PC and brain regions controlling affective states such as the orbitofrontal cortex, the nucleus accumbens or amygdala. In fact, there is compelling literature demonstrating the participation of these brain areas in olfactory processes [56,57,58]. In line with this idea, recent observations in humans highlight that the state-dependent enhancement of endocannabinoid levels changes dietary choices toward high-energy food items. Interestingly, this phenomenon was related to an increase in odor responses in the PC [39].

## 6. Conclusions

Growing evidence has revealed that the ECS modulates direct olfactory processes such as odor sensitivity or olfactory learning and memory. Across different brain olfactory areas, the ECS appears to play an essential role in the control of synaptic transmission and plasticity, but also in the regulation of vital behaviors that depend on olfaction, such as the feeding state of the individual (Figure 1; [59]). However, the physiological impact of the endocannabinoid-mediated plasticity, the contribution of CB1 receptors during other olfactory-dependent behaviors, and the contribution of each olfactory brain region (e.g., OE, MOB, PC) during specific behaviors remain to be elucidated. Furthermore, the role of other components of the ECS in olfactory processes is less clear, such as the role of CB2 receptors, which are described to be present in the OE [27]. This highlights the importance of continuing with this exciting line of research. To the same extent, there is a lack of direct evidence about the participation of other olfactory-related structures in cannabinoid-mediated effects, for example, the olfactory tubercle (OT). In fact, the OT is a target of different hormones and local modulators regulating feeding behavior and motivation [48]. Thus, it is reasonable to think about a potential cross-link between cannabinoid signaling and hormonal regulation in olfactory related behaviors taking place in the OT.

Besides the studies regarding the functions of the ECS in primary olfactory structures, it is important to take into account that CB1 receptors are present and modulate associated olfactory areas (i.e., amygdala, orbitofrontal cortex, hippocampus or periaqueductal gray; [41,42,43,44]), suggesting that olfactory processing that involves the control of different brain structures might also be modulated by the ECS. In humans, the main psychoactive compound of cannabis has been shown to induce an increase in olfactory perception and disturbs odor discrimination and pleasantness [60,61,62]. Furthermore, a recent study shows that cannabis consumption could also affect other neurotransmitter systems in olfactory structures [63]. One of the main characteristics of the CB1 receptor’s activity is its bimodal activity: the cell-type of where it is expressed can lead to opposite effects (e.g., CB1 in GABAergic cells promotes satiety while in glutamatergic cells it induces hunger) [64]. This bimodal action could also be present in olfactory processes considering the pattern of expression of the CB1 receptors, but future research is needed to clarify this crucial point.

The interconnectivity between olfactory areas, together with the tight ECS-control of various types of cells and subcellular locations, makes the determination of the different roles of CB1 receptors in the olfactory system very complex and challenging. A better understanding of such interactions will result not only in a significant advance for neuroscience, but could also lead to novel human-based studies targeting specific populations. Interestingly, alterations of ECS functioning have been shown to contribute to the development of neurological and neuropsychiatric disorders in which loss of smell represents the early stages of the disease [65,66,67,68]. All of this information could provide the rationale to propose a combined use of olfactory manipulations with ECS-based pharmacotherapy to potentially treat pathological conditions.

## Figures and Tables

**Figure 1 genes-11-00431-f001:**
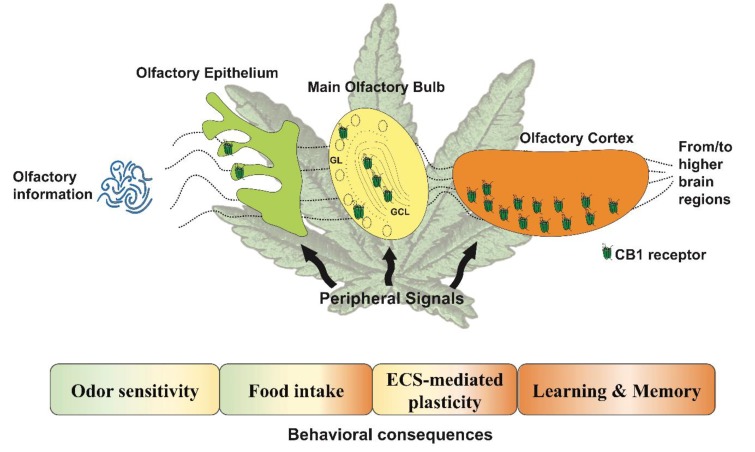
Main functions regulated by the cannabinoid signaling described in the primary olfactory-dependent structures. The endocannabinoid system controls various functions that depend on the olfactory areas involved. The colors in the boxes indicate which structure is involved in the associated function. The scheme also shows how the different localization of the cannabinoid type-1 (CB1) receptor potentially modulates the flow of olfactory information from early sensory coding to more complex computations, and is modulated by peripheral signals, resulting in behavioral outputs. GL: glomerular layer; GCL: granular cell layer; the olfactory cortex includes both the anterior olfactory nucleus and the piriform cortex, ECS: endocannabinoid system.

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
