# Peer review of "Cannabinoid Control of Olfactory Processes: The Where Matters"

_genes, 2020, doi:10.3390/genes11040431_

Round 1

Reviewer 1 Report

In their Review, Geoffrey Terral and colleagues describe the involvement of the endocannabinoid (eCB)-mediated system in regulating three main olfactory-related structures: the olfactory epithelium, the main olfactory bulb, and the piriform cortex.

The role of the eCB system in olfactory related processes is still an emerging field of study. This work clearly summarizes the most relevant current findings in this topic.

I only have minor points.

1) The Authors describe the function of Cannabinoid receptor type 1 (CB1) as a major receptor for (endo)cannabinoids in the three main olfactory-related structures. However, Hutch and colleagues (Hutch CR, Hillard CJ, Jia C, Hegg CC. An endocannabinoid system is present in the mouse olfactory epithelium but does not modulate olfaction. Neuroscience. 2015;300:539–553. doi:10.1016/j.neuroscience.2015.05.056) reported the presence of Cannabinoid receptor type 2 (CB2) in the mouse olfactory epithelium. By revising the literature, it emerges that the role of CB2R in olfaction is even less clear than the one of CB1R. Nevertheless, the authors should mention the potential implication of CB2R expression.  Alternatively, the title of the review should indicate that the manuscript focuses selectively on CB1R

2) A growing body of evidence points at the olfactory tubercle as an important olfactory structure, especially in motivated behavior. Since this brain structure is mainly populated by Medium Spiny neurons, whose activity is well known to be modulated by the eCB system. Is there any specific reason that prompted the authors to not consider the olfactory tubercle in their review? Is it known whether the eCB system plays a role in regulating the activity of this brain region?

3) Figure 1 of the manuscript:

  • in the main text, the authors describe the involvement of CB1R in the Piriform Cortex without mentioning the Olfactory Cortex. However, figure 1 they refer only to the olfactory cortex. The piriform cortex should be also included.
  • In my opinion, the schematic of the behavioral consequences is misleading. There is no apparent link between the involvement of the eCB system in seizures and olfaction. I think it would be more appropriate to represent here the olfactory related behaviors that are regulated by eCBs.

My understanding is that findings showing that the Piriform Cortex can undergo epileptic seizures helped to establish the role of the eCB system in this area, but this role is not related to olfaction per-se.

  • In the figure legend are reported some acronyms which are not used in the figure itself.

Reviewer 2 Report

The current manuscript by Terral et al. provides a nice summary on the role of the endocannabinoid system in controlling the function of olfactory pathways. This is an important convergence of two topics, not often reviewed, which should be of relevance to numerous readers. I have only a few minor suggestions below which I hope may improve the manuscript:

  1. The endocannabinoid system: A general overview

Line 39: “…activation of metabotropic acetylcholine and glutamate receptors.”

- While this line exists to simply state a general pathway of endocannabinoid signalling, it would be more accurate to specifically state that metabotropic, Gq-coupled receptors (i.e. M1/M3 and mGluR1/5) are involved.

- Caution should also be taken in stating that both these mechanisms can induce endocannabinoids by increasing postsynaptic intracellular calcium levels. While Group 1 mGluR activation can mobilize Ca2+ from internal stores (which can facilitate Ca2+-dependent release during DSI/DSE), it primarily leads to endocannabinoid induction via a distinct Ca2+-independent mechanism (see Maejima et al., 2001, 2005)). Thus, the above line should be slightly modified to suggest two distinct mechanisms of endocannabinoid release.

  1. Role of the endocannabinoid system in olfactory circuits

 - Given that a predominant subjective effect of cannabis intoxication is altered olfactory perception, it might be interesting to comment on the relative levels of CB1 receptor expression compared to other well studied regions (if the data exists).

  1. The endocannabinoid system in Olfactory Sensory Neurons

- One of the most intriguing points in this section is the relationship between satiety state and olfactory perception. Perhaps it might be beneficial to briefly describe the circuitry linking these two systems (i.e. how afferent feeding inputs synapse onto the PC/MOB/OSN).

  1. The endocannabinoid system in the Olfactory Bulb

Line 101: “…express by…”

- should be corrected to “expressed on” or “Since CB1 receptors are expressed by CFF fibers…”

  1. The endocannabinoid system in the Piriform Cortex

Line 111: The PC is a brain area capable of generating epileptiform activity

- this is quite a modest statement here. More emphasis could be placed on the piriform cortex being a unique region sensitive to epileptiform activity. Indeed, it is one of the most susceptible of all regions to seizures. Is this property due to its unique anatomy, or is there possibly something unique about the endocannabinoid system?

Line 125-126: Despite the possible post-synaptic CB1 receptors localization, it is not yet understood how CB1 receptor activation allows controlling GABAergic conductance.

- there is now evidence suggesting that 2-AG can have a direct postsynaptic action on neurons (via a reduction of A-type potassium channels)(see Gantz and Bean, 2017). Any chance this may mediate the effects observed by Ghosh et al. (34)? The latter study suggests a PKA-dependent mechanism by cannabinoids, which is known to modulate KA. This would need to be looked into and is optional to discuss.

- In both Section 4 and 5, the endocannabinoid system appears to primarily drive GABAergic transmission in the main olfactory bulb and piriform cortex. Is there possibly a reason why it is so specific to GABA? Perhaps a brief comment on where CB1R is specifically expressed (i.e. GABA vs. glutamate) may help.

- Related to the above, work from the authors’ own lab has shown that CB1 receptors on GABAergic vs glutamatergic synapses in the forebrain may have opposing effects on food intake (see Bellocchio et al., 2010). Does a similar segregation possibly exist in olfactory pathways for bimodal control of feeding? This would be an interesting addition to the manuscript if so.

- This section highlights the important role of the aPC in regulating aversive/appetitive odor memory. Again, it might be worth briefly describing the circuitry linking appetitive/aversive stimuli to olfaction.

  1. Conclusion

In addition to pointing out that changes in the ECS system and olfactory pathways (i.e. anosmia) may precede certain neuropsychiatric disorders, it would be interesting to highlight any potential pharmacotherapies/treatments being developed (or the clinical use of THC) to target such diseases. 

REFERENCES

Bellocchio, L., Lafenêtre, P., Cannich, A., Cota, D., Puente, N., Grandes, P., Chaouloff, F., Piazza, P.V., and Marsicano, G. (2010). Bimodal control of stimulated food intake by the endocannabinoid system. Nat. Neurosci. 13, 281–283.

Gantz, S.C., and Bean, B.P. (2017). Cell-Autonomous Excitation of Midbrain Dopamine Neurons by Endocannabinoid-Dependent Lipid Signaling. Neuron.

Maejima, T., Hashimoto, K., Yoshida, T., Aiba, A., and Kano, M. (2001). Presynaptic inhibition caused by retrograde signal from metabotropic glutamate to cannabinoid receptors. Neuron 31, 463–475.

Maejima, T., Oka, S., Hashimotodani, Y., Ohno-Shosaku, T., Aiba, A., Wu, D., Waku, K., Sugiura, T., and Kano, M. (2005). Synaptically driven endocannabinoid release requires Ca2+- assisted metabotropic glutamate receptor subtype 1 to phospholipase C β4 signaling cascade in the cerebellum. J. Neurosci.
